# OpenReview forum: "Reinforcing the Diffusion Chain of Lateral Thought with Diffusion Language Models"
_NeurIPS.cc/2025/Conference — NeurIPS 2025 poster_

### Official Review · Reviewer_bbyf · 2025-06-22

**Clarity:** 3
**Significance:** 3
**Originality:** 3
**Rating:** 4
**Confidence:** 3

**Summary:**

This paper introduces DCoLT, a new framework for reasoning in Diffusion Language Models (DLMs) that follows lateral thinking instead of traditional step-by-step reasoning. It uses reinforcement learning to train all intermediate steps in the reverse diffusion process, with an Unmasking Policy Module (UPM) using a Plackett-Luce model.

**Questions:**

1. Have authors considered integrating auxiliary rewards alongside outcome-based RL?

2. Provide concrete examples of failure cases

3. Can you provide theoretical analysis of when and why DCoLT should outperform traditional approaches?

4. How does the computational cost compare to autoregressive methods with similar performance?

5. Can you demonstrate that the "lateral thinking" behavior is consistent and beneficial across a broader range of problems?

6. How does the approach scale to larger models and longer sequences?

**Ethical Concerns:**

["NO or VERY MINOR ethics concerns only"]

**Final Justification:**

The authors have addressed most of my concerns in their rebuttals, and I believe a score of 4 is appropriate for this paper.

**Limitations:**

yes

**Quality:**

3

**Strengths And Weaknesses:**

I'll provide a comprehensive review of this paper on "Reinforcing the Diffusion Chain of Lateral Thought with Diffusion Language Models."

## Summary

This paper introduces DCoLT (Diffusion Chain of Lateral Thought), a novel reasoning framework for diffusion language models that treats intermediate diffusion steps as "thinking" actions and optimizes the entire reasoning trajectory using outcome-based reinforcement learning. The authors implement DCoLT on two diffusion language models (SEDD and LLaDA) and demonstrate improvements on math and code generation tasks.

## Strengths

### Quality
- **Strong experimental validation**: The paper provides comprehensive experiments across multiple tasks (GSM8K, MATH, MBPP, HumanEval) and compares against relevant baselines including both diffusion models and autoregressive models.
- **Solid technical implementation**: The authors properly adapt RL training to two different types of diffusion models (continuous-time SEDD and discrete-time LLaDA), showing technical depth.
- **Thorough analysis**: The paper includes helpful visualizations of the generation process and ablation studies examining key components like the Unmasking Policy Module.

### Clarity
- **Well-structured presentation**: The paper is generally well-written with clear motivation and methodology sections.
- **Good visual aids**: Figure 1 effectively illustrates the key differences between CoT and DCoLT, and the appendix provides helpful examples of the generation process.
- **Comprehensive details**: The extensive appendix provides implementation details necessary for reproduction.

### Significance
- **Novel perspective on reasoning**: The concept of "lateral thinking" in diffusion models is interesting and provides a fresh angle on reasoning in language models.
- **Competitive results**: DCoLT achieves strong performance, particularly on LLaDA where it shows substantial improvements (+9.8% on GSM8K, +5.7% on MATH).
- **Data efficiency**: The approach demonstrates good data efficiency compared to methods requiring much larger proprietary datasets.

## Weaknesses

### Technical Concerns
1. **Limited theoretical justification**: The paper lacks theoretical analysis of why treating diffusion steps as reasoning should work better than traditional approaches. The "lateral thinking" metaphor, while intuitive, needs stronger theoretical grounding.

2. **Unclear advantage over autoregressive models**: While the results are competitive, it's not entirely clear why the added complexity of diffusion models is justified when autoregressive models with CoT can achieve similar or better performance with much simpler training.

3. **Reward design limitations**: The approach is limited to tasks with verifiable rewards (math/code), which significantly constrains its applicability. The authors acknowledge this but don't provide a clear path forward.

### Experimental Limitations
1. **Limited model scales**: Experiments are conducted on relatively small models (400M SEDD, 8B LLaDA). It's unclear how the approach scales to larger models where autoregressive methods have shown dramatic improvements.

2. **Computational efficiency concerns**: The paper doesn't thoroughly analyze the computational overhead of the multi-step diffusion process compared to autoregressive generation, which is crucial for practical adoption.

3. **Limited baseline comparisons**: While the paper compares to some diffusion models, it would benefit from more comprehensive comparisons with recent strong autoregressive reasoning methods.

### Methodological Issues
1. **Unclear generalization**: The "lateral thinking" concept is demonstrated mainly through examples, but it's unclear how consistently this emerges across different problems and whether it's truly beneficial or just a different path to the same solution.

2. **Evaluation metrics**: The paper focuses primarily on accuracy metrics but doesn't analyze other important aspects like reasoning quality, interpretability, or failure modes.

---

> ### Author Rebuttal · Authors · 2025-07-31
>
> **Q1: Have authors considered integrating auxiliary rewards alongside outcome-based RL?**
>
> In this work, we focus exclusively on outcome-based rewards to ensure a fair comparison with existing models. We agree that incorporating auxiliary rewards, such as step-level correctness, could potentially enhance performances. We will explore it in future work.
>
> **Q2: Provide concrete examples of failure cases**
>
> Thank you for the suggestion. We provide an example in Fig. 12(b), where DCoLT generates incorrect reasoning due to a miscalculation in an intermediate step. This type of error often arises when complex problems are not adequately decomposed into trackable steps.
>
> As shown in Fig. 12(b), the failure case was corrected by increasing the generation length to $L=512$, which enables finer-grained decomposition of reasoning steps. This is also supported by the improved performance with longer generations shown in Tab 4 in the submission. We will add more failure cases and show how they can be remedied by the DCoLT.
>
> **Q3: Can you provide theoretical analysis of when and why DCoLT should outperform traditional approaches?**
>
> One of DCoLT's unique features is it learns the order of generated tokens based on the Plackett–Luce model, grounded in Luce’s Axiom of Choice [a] and Random Utility Model [b]. This enables the model to explicitly learn an optimal token generation order, rather than sticking to a fixed left-to-right order as in autoregressive models.
>
> This flexibility is particularly advantageous in tasks where the natural problem-solving order differs from the left-to-right text order, such as in Sudoku. For example, as shown in Fig. 3, a natural order to complete the task is to fill in easier cells first, and then use them as the context to infer more difficult ones.
>
>
> [a] Mollica C, et al. Remarkable properties for diagnostics and inference of ranking data modelling. British Journal of Mathematical and Statistical Psychology, 2022.
>
> [b] Ma J, et al. Learning-to-rank with partitioned preference: Fast estimation for the Plackett-Luce model. International Conference on Artificial Intelligence and Statistics, 2021.
>
> **Q4: How does the computational cost compare to autoregressive methods with similar performance?**
>
> We provide an inference-time comparison between DCoLT and a recent autoregressive reasoning model, Deepseek-R1-Distill-Llama-8B. Specifically, we randomly sample 500 problems from GSM8K and report the average wall-clock inference time required to complete a full response. As shown in Tab. I, DeepSeek-R1-Distill-Llama-8B exhibits higher inference latency. In contrast, DCoLT requires less inference time, with even better accuracy.
>
> **Table I**: Computation cost to solve one problem in GSM8K. The inference latency is averaged on 500 problems.
>
> | Model                        | Accuracy | Inference Latency (s) |
> | ---------------------------- | -------- | --------------------- |
> | LLaDA + DCoLT 8B             | 88.1%    | 6.34                  |
> | DeepSeek-R1-Distill-Llama-8B | 77.3%    | 10.80                 |
>
> **Q5: Can you demonstrate that the "lateral thinking" behavior is consistent and beneficial across a broader range of problems?**
>
> DCoLT demonstrates consistent improvements across both math and code generation tasks, as shown in our results on MATH, GSM8K, MBPP and HumanEval. These various domains involve complex multi-step reasoning, and the observed gains suggest that lateral exploration during reasoning is broadly beneficial.
>
> While this work focuses on tasks with verifiable answers, we believe that the principles behind DCoLT generalize to broader settings. In future, we will extend DCoLT to open domains(e.g., scientific inquiry and creative generation) with trained reward models to guide lateral reasoning where verifiable answers are not available.
>
> **Q6: How does the approach scale to larger models and longer sequences?**
>
> During RL training, LLaDA+DCoLT that is optimized with a generation length of $L=256$ has already outperformed LLaDA and other RL algorithms (e.g., d1) under the same length. As shown in Tab.4, we showed that scaling the generation length further improves the model performance.
>
> Regarding scalability with increasing model sizes, our approach demonstrates consistent gains on both smaller SEDD models (400M) and larger LLaDA models (8B). To our best knowledge, LLaDA 8B model is the largest open-source diffusion language model. We will test the ability of DCoLT scaling to larger DLMs once they are available.

---

> > ### Comment · Reviewer_bbyf · 2025-08-04
> >
> > Thank you for your comments, I will maintain my score.

---

### Official Review · Reviewer_h4x9 · 2025-06-28

**Clarity:** 3
**Significance:** 3
**Originality:** 3
**Rating:** 5
**Confidence:** 3

**Summary:**

This paper introduces the Diffusion Chain of Lateral Thought, a reasoning framework for diffusion language models that treats reverse diffusion steps as thinking actions and optimizes them via outcome-based RL. Unlike linear CoT methods, DCoLT enables bidirectional, non-linear reasoning without strict grammatical constraints on intermediate steps. Experiments show that DCoLT-reinforced models outperform SFT and RL baselines on math and code generation tasks.

**Questions:**

1. How does DCoLT's RL approach (e.g., GRPO) compare to other RL algorithms (e.g., PPO) in terms of training stability and performance?

2. The paper shows longer generations improve accuracy, but what is the optimal length for tasks with different degrees of difficulty, and how does it balance computation and model performance?

**Ethical Concerns:**

["NO or VERY MINOR ethics concerns only"]

**Final Justification:**

I am satisfied with the rebuttal.

**Limitations:**

Yes

**Paper Formatting Concerns:**

Nil

**Quality:**

3

**Strengths And Weaknesses:**

Strengths:
1. DCoLT leverages reverse diffusion to model lateral thinking, allowing non-linear, format-free reasoning that aligns with human cognitive patterns, which is not supported by causal CoT.
2. Using only public data and 16 H800 GPUs, DCoLT-reinforced DLMs achieve state-of-the-art results, outperforming models trained with proprietary data.
3. Supporting both continuous-time and discrete-time diffusion language models, DCoLT demonstrates versatility across different diffusion paradigms.

Weaknesses:
1. Performance could be improved with more data and compute, and prior works with proprietary data show better results.
2. DCoLT is validated only on tasks with clear reward functions e.g., math and coding, lacking evaluation on open-ended natural language tasks.

---

> ### Author Rebuttal · Authors · 2025-07-31
>
> **Response to Strengths And Weaknesses**
>
> We thank the reviewers for acknowledging our contributions in modeling lateral thinking through reverse diffusion, achieving competitive results using only public data and moderate compute, and demonstrating versatility across different diffusion language models.
>
> We agree that performance can be further improved with access to larger datasets and increased computational resources. We also recognize the importance of extending DCoLT to open-ended natural language tasks. To this end, we are exploring the integration of learned reward models to tackle these tasks, and will scale DCoLT in future works to further showcase the generality and potential of DCoLT across diverse reasoning challenges.
>
> **Q1: How does DCoLT's RL approach (e.g., GRPO) compare to other RL algorithms (e.g., PPO) in terms of training stability and performance?**
>
> In the paper, we adopt GRPO in DCoLT for a fair comparison with existing models (e.g., d1). Prior studies [a] have noted that PPO could suffer an unstable training process to learn a value network particularly in long reasoning chains. In contrast, GRPO completely eliminates the need for learning a value network, thereby demonstrating more stable training with better performances as evidenced by existing works [b].
>
>
>
> [a] Yuan Y, et al. What’s Behind PPO’s Collapse in Long-CoT? Value Optimization Holds the Secret. 2025.
>
> [b] Shao Z, et al. Deepseekmath: Pushing the limits of mathematical reasoning in open language models. 2024.
>
> **Q2: What is the optimal length for tasks with different degrees of difficulty, and how does it balance computation and model performance?**
>
> We evaluate LLaDA+DCoLT 8B on MATH subsets grouped by different difficulty levels, following the original dataset annotations[c].  As shown in Tab. I, the accuracy on level-1 problems saturates with length 256, suggesting that shorter generations are sufficient for simpler questions. In contrast, the performance on harder problems (levels 2–5) continues to improve with longer generations, indicating that complex reasoning benefits from longer responses. This complies with our intuition that optimal generation length is dependent on difficulty levels — shorter for easier problems, and longer for more difficult ones. This also reflects a trade-off between computational cost and model performance, allowing generation length to be adapted to varying difficulty levels.
>
> **Table I**: Accuracy of LLaDA+DCoLT 8B on MATH subsets across difficulty levels (5: the hardest and 1: the easiest) and generation lengths.
>
> | Length | Difficulty Level 1 | Difficulty Level 2 | Difficulty Level 3 | Difficulty Level 4 | Difficulty Level 5 |
> | ------ | ------------------ | ------------------ | ------------------ | ------------------ | ------------------ |
> | 128    | 80.8%              | 61.0%              | 45.6%              | 29.5%              | 13.2%              |
> | 256    | 83.3%              | 65.1%              | 52.3%              | 35.9%              | 18.7%              |
> | 384    | 82.4%              | 66.6%              | 54.9%              | 39.0%              | 20.2%              |
> | 512    | 82.6%              | 69.7%              | 56.9%              | 40.2%              | 21.5%              |
>
>
>
> [c] Hendrycks D, et al. Measuring Mathematical Problem Solving With the MATH Dataset. NeurIPS 2021.

---

> > ### Author Response · Authors · 2025-08-06
> >
> > Dear Reviewer h4x9，
> >
> > Thank you for your initial feedback on our work. We appreciate the time you’ve taken to evaluate our paper and have carefully addressed your questions in our rebuttal. To ensure all concerns are fully resolved before the discussion deadline (**Aug 8 AoE**), could you kindly let us know if you have any remaining concerns or need clarification on any points? We’re happy to provide additional details.

---

> > ### Comment · Reviewer_h4x9 · 2025-08-07
> >
> > Thanks for the rebuttal. I remain positive with the paper.

---

### Official Review · Reviewer_P9Mj · 2025-07-01

**Clarity:** 3
**Significance:** 3
**Originality:** 3
**Rating:** 4
**Confidence:** 4

**Summary:**

The paper proposes an online reinforcement learning algorithm called DCoLT for finetuning masked diffusion models on reasoning and code. The method uses group statistics to estimate the advantage function without using a explicit value function. The RL objective is similar to GRPO but it incorporates all the steps from the diffusion trajectory (unlike other works like d1 that only use a single step estimate). The paper also introduces an additional “unmasking” head for selecting the tokens to unmask, and show that this in itself provides significant gains in performance even if the rest of the model is frozen. The results are quite strong showing improvement over existing SFT and other RL baselines.

**Questions:**

1. Based on Appendix C.3, it appears that d1 trains separate models for GSM8K and MATH, each using its own corresponding training set. However, DCoLT seems to combine both datasets during training. This may not be a fair comparison. Can you please clarify whether the same training data is used for both models?

2. I may be misunderstanding, but it seems like d1 only finetunes LoRA adapters, whereas DCoLT does full parameter updates, making the results difficult to compare.

3. The term “nonlinear generation” is ambiguous. For instance, in line 131, I don’t quite understand what it means. If the intent for it is to mean “non-left-to-right,” then it might be better to write non-left-to-right or something similar.

4. Could you comment on the coherence of the complete answer sequence when the answer is correct? I realize that a final-answer-based 0/1 reward may not encourage, or perhaps even discourage, coherent text sequence in the middle of the answer. It will be nice to present some concrete analysis on this front.

5. Please provide more details regarding the final answer-based reward for the question-answering datasets. Specifically, what is the filter used to extract the final answer for GSM8K-Aug, GSM8K, and MATH datasets for reward generation?

6. Do you train the model to predict <eos> tokens? In Figure 8, in step 64, I see that there are <eos> tokens. Are these predicted by the model, or are they inserted automatically after the model predicts the first <|eot_id|>?

7. Have you tried DCoLT on Dream 7B? It would make the paper stronger if the method shows similar gains across the two models.

8. Since RL training is generally unstable, it would be good if you could provide average reward vs training progress curves.

**Ethical Concerns:**

["NO or VERY MINOR ethics concerns only"]

**Final Justification:**

I believe that there is scope to further improve the paper by providing simpler baselines, such as diffusion SFT training with rejection sampling. However, by providing an apples-to-apples comparison (using LoRA instead of full fine-tuning) with the only other diffusion+RL baseline in the literature for text diffusion models, the authors' response somewhat addresses my concern.

**Limitations:**

yes

**Quality:**

2

**Strengths And Weaknesses:**

**Clarity**: The paper is well written and easy to follow. There is a good amount of implementation-related details in the paper and the appendix, suggesting that the results should be easily reproducible.

**Quality**: The baselines are a bit weak. Specifically, the only Diffusion+RL-based baseline is d1-LLaDA. But it is unclear if d1 can be compared directly with the proposed approach (see questions 1, and 2) below.

**Significance**: The work could be quite useful for the community, as it demonstrates that full-trajectory based RL performs reasonably well for masked diffusion models.

**Originality**: The work is original because it is the first to successfully apply trajectory-level RL for the masked diffusion model in the text domain.

---

> ### Author Rebuttal · Authors · 2025-07-31
>
> **Q1 & Q2:  Based on Appendix C.3, it appears that d1 trains separate models for GSM8K and MATH, each using its own corresponding training set. However, DCoLT seems to combine both datasets during training. This may not be a fair comparison. Can you please clarify whether the same training data is used for both models? I may be misunderstanding, but it seems like d1 only finetunes LoRA adapters, whereas DCoLT does full parameter updates, making the results difficult to compare.**
>
> A direct comparison with d1 using the same LoRA structure and dataset still reveals the superior performance by DCoLT. Specifically, we train DCoLT on GSM8K only, using LoRA with rank $r=128$ and scaling factor $\alpha=64$. During inference, we set the generation length to 256, the number of diffusion steps to 128, and the block length to 32 — mirroring d1’s configuration. In the same setting, DCoLT with LoRA achieves an accuracy of 84.7%, which outperforms d1 with both diffu-GRPO (79.8%) and d1-LLaDA (81.1%) setups, as shown in Tab. I. These results indicate that DCoLT still performs better than d1 with the same LoRA structure and the training data.  We will add it in the final revision.
>
> **Table I**: Comparison between d1 and DColT with LoRA on GSM8K. Both methods only use the GSM8K to train the LLaDA model with their respective RL approach.
>
> | Model                       | Accuracy |
> | --------------------------- | -------- |
> | LLaDA 8B                    | 78.3%    |
> | +diffu-GRPO                 | 81.9%    |
> | d1-LLaDA (SFT + diffu-GRPO) | 82.1%    |
> | **LLaDA + DCoLT 8B (LoRA)** | 84.7%    |
>
> **Q3: The term “nonlinear generation” is ambiguous. For instance, in line 131, I don’t quite understand what it means. If the intent for it is to mean “non-left-to-right,” then it might be better to write non-left-to-right or something similar.**
>
> In our paper, "nonlinear generation" intends to mean "non-left-to-right". We will clarify this in the revision.
>
> **Q4: Could you comment on the coherence of the complete answer sequence when the answer is correct? I realize that a final-answer-based 0/1 reward may not encourage, or perhaps even discourage, coherent text sequence in the middle of the answer. It will be nice to present some concrete analysis on this front.**
>
> Although DCoLT is trained using only final-answer-based 0/1 rewards, we observe that it often produces coherent text sequences as shown in Fig. 8-9 of Appendix B. To further quantify this, we decompose the generated text responses on GSM8K into individual reasoning steps and evaluate their text coherency quality using a process reward model (Qwen2.5-Math-PRM-7B [a]). On correctly answered examples, LLaDA+DCoLT achieves an average step-wise reward of 0.96 (very close to the maximum reward of 1.0), better than LLaDA with an average reward of 0.94. This suggests that DCoLT produces not only correct answers but also coherent text sequence.
>
> The reason can be that: unlike auto-regressive models that are pretrained to predict next token from readable text prefix, DLMs are pretrained [b] to recover coherent text from randomly masked inputs that are incomplete and unreadable. This training strategy endows DLMs with the unique robustness to generate coherent text outputs from corrupted text sequences amid the reverse diffusion process. This trained ability of DLMs allows the outcome-based RL to maintain the coherence of the final output sequences without relying on explicit rewards over intermediate diffusion steps to recover masked tokens.
>
>
>
> [a] Zhang Z, et al. The Lessons of Developing Process Reward Models in Mathematical Reasoning. 2025.
>
> [b] Nie S, et al. Large language diffusion models. 2025.
>
> **Q5: Please provide more details regarding the final answer-based reward for the question-answering datasets. Specifically, what is the filter used to extract the final answer for GSM8K-Aug, GSM8K, and MATH datasets for reward generation?**
>
> For GSM8K-Aug experiments with SEDD, we follow the approach used in DoT [c]: if the model outputs contain `####`, we extract the number that follows as the predicted answer. Example responses following this pattern are provided in Fig. 6.
>
> For GSM8K and MATH experiments with LLaDA, we follow DeepseekMath [d] and require the final answer to be put in the right response format (e.g., be enclosed in a LaTeX-style box `\box{}`). During reward computation, we extract the content inside the box and attempt to parse it into a number or symbolic expression. Then, we compare it against the ground-truth answer using symbolic math equivalence checks, following the evaluation code from the MATH dataset by Hendrycks et al [e]. Examples of accepted answer formats are shown in Fig. 8 and Fig. 9 of Appendix B.
>
>
>
> [c] Ye J, et al. Diffusion of thought: Chain-of-thought reasoning in diffusion language models. NeurIPS 2024.
>
> [d] Shao Z, et al. Deepseekmath: Pushing the limits of mathematical reasoning in open language models. 2024.
>
> [e] Hendrycks D, et al. Measuring Mathematical Problem Solving With the MATH Dataset. NeurIPS 2021.
>
> **Q6: Do you train the model to predict [object Object] tokens? In Figure 8, in step 64, I see that there are [object Object] tokens. Are these predicted by the model, or are they inserted automatically after the model predicts the first <|eot_id|>?**
>
> The `<eos>` tokens are predicted by the model. These predictions are considered as part of the trajectory to reinforce during training.
>
> **Q7: Have you tried DCoLT on Dream 7B? It would make the paper stronger if the method shows similar gains across the two models.**
>
> We appreciate the reviewer’s suggestion and conduct an additional experiment applying DCoLT to Dream 7B on GSM8K, using 64 denoising steps and a generation length of 256. As shown in Tab. II, DCoLT improves Dream 7B's accuracy to 80.53%, representing a +30.42% gain over the original model — comparable to the improvement observed on LLaDA.
>
> Due to limited time during the rebuttal period, we have not yet conducted a experiment with 256 denoising steps, but we plan to include those results in the revised version of the paper.
>
> **Table II**: Experimental results by applying DCoLT to Dream 7B on GSM8K with 64 denoising steps.
>
> |      Models      |     Accuracy     |
> | :--------------: | :--------------: |
> |     Dream 7B     |      50.11%      |
> | Dream + DCoLT 7B | 80.53% (+30.42%) |
>
> **Q8: Since RL training is generally unstable, it would be good if you could provide average reward vs training progress curves.**
>
> Thanks for your constructive advice. During training, our model exhibits a stable increase in reward. Due to formatting constraints, we are unable to include these figures in the rebuttal. However, we will include training curves in the revised version to illustrate the training process.

---

> > ### Comment · Reviewer_P9Mj · 2025-08-05
> > **Thanks for answering my questions.**
> >
> > Thanks for answering my questions and providing additional results with LoRA. In the subsequent revisions, please mention that the d1 baseline uses LoRA and not full fine-tuning. I have no further questions.

---

### Official Review · Reviewer_pRnQ · 2025-07-04

**Clarity:** 2
**Significance:** 2
**Originality:** 3
**Rating:** 3
**Confidence:** 3

**Summary:**

This paper introduces a novel method, diffusion chain of lateral thought, which uses reinforcement learning with final answer only rewards to enhance reasoning ability of  diffusion language models. Authors develop methods for both continuous and discrete time diffusion models. They claim this could have bidirectional and format free reasoning abilities. Authors also implement experiments with SEDD and LLaDA with DCoLT on math, coding and sudoku benchmarks, and present competitive or better performance.

**Questions:**

The method only trained the model using final-answer reward signals, with no supervision or constraint on the intermediate diffusion steps. This setup has been shown to cause unreadable intermediate steps in autoregressive models (e.g. DeepSeek-R1), and this is a reason why they have an additional SFT process. Despite this, attached examples in this paper show semantically meaningful and readable CoT like reasoning. I’m curious why the case is different with autoregressive models and outcome-based RL on diffusion language models that lead to interpretable intermediate reasoning.

**Ethical Concerns:**

["NO or VERY MINOR ethics concerns only"]

**Final Justification:**

Please refer to my original comments and my response to the authors feedback during the rebuttal.

**Limitations:**

See weaknesses.

**Quality:**

2

**Strengths And Weaknesses:**

Strengths:
* It tries to simulate lateral, human-like thinking, where key concepts emerge early and are refined over time. This is a clear advancement over traditional Chain-of-Thought in autoregression models, because autoregressive models cannot modify generated tokens.
* Authors thoughtfully implement DCoLT on both continuous time and discrete time diffusion language models. This highlights the adaptability across different model architectures.
* This method leverages final answer only rewards, which is a practical strength, as high quality CoT annotations are expensive and limited.
* The method achieves better or competitive results on math, coding and sudoku benchmarks. It performs even better than its autoregressive counterpart in some tasks.

Weaknesses:
* Some claims in this paper are problematic, e.g. for the generation length, I don’t think it plays an important role here. The performances with length from 256-512 are very close.
* I’m not sure whether the method really optimizes the entire reverse process jointly (line 173-174), because in Algorithm1, the optimization is processed from 1 to n, which means getting loss for each denoising step separately.

---

> ### Author Rebuttal · Authors · 2025-07-31
>
> **W1: Some claims in this paper are problematic, e.g. for the generation length, I don’t think it plays an important role here. The performances with length from 256-512 are very close.**
>
> During RL training, LLaDA+DCoLT 8B is only optimized with a generation length of $L=256$, and it already outperforms LLaDA and other RL algorithms (e.g., d1) under the same length setting. In the submission, inspired by prior works on auto-regressive models, we also investigate the effect of generation length on performances in Tab. 4 as well.
>
> Particularly, although the overall performance with varying lengths may be close, when they are applied to MATH subsets with various difficulty levels,  LLaDA+DCoLT 8B with longer generations could perform much better on harder problems than that with shorter ones as shown in Tab. I. For example, on the hardest category with difficulty level 5, DCoLT improves the accuracy from 13.2% with a generation length of 128 to 21.5% with that of 512, a remarkable gain of 8.3% in accuracy.
>
> **Table I**: Accuracy of LLaDA+DCoLT 8B on MATH subsets across difficulty levels (5: the hardest and 1: the easiest) and generation lengths.
>
> | Length | Difficulty Level 1 | Difficulty Level 2 | Difficulty Level 3 | Difficulty Level 4 | Difficulty Level 5 |
> | ------ | ------------------ | ------------------ | ------------------ | ------------------ | ------------------ |
> | 128    | 80.8%              | 61.0%              | 45.6%              | 29.5%              | 13.2%              |
> | 256    | 83.3%              | 65.1%              | 52.3%              | 35.9%              | 18.7%              |
> | 384    | 82.4%              | 66.6%              | 54.9%              | 39.0%              | 20.2%              |
> | 512    | 82.6%              | 69.7%              | 56.9%              | 40.2%              | 21.5%              |
>
>
> **W2: I’m not sure whether the method really optimizes the entire reverse process jointly (line 173-174), because in Algorithm1, the optimization is processed from 1 to n, which means getting loss for each denoising step separately.**
>
> We compute and accumulate the losses for **all** the denoising steps in the entire reverse diffusion process, and sum them up to update the model parameters. Note that the network weights were not updated **until** the losses from all the steps are accumulated to optimize the entire reverse process.
>
> Moreover, each one-diffusion-step gradient accumulated in Line 26-28 of Algorithm 1 fulfills a memory-efficient strategy. As described in Line 149, we release the computational graph after each step to reduce memory consumption, which still optimizes the entire reverse process jointly with the accumulated gradient in a memory-efficient fashion.
>
> **Q: The method only trained the model using final-answer reward signals, with no supervision or constraint on the intermediate diffusion steps. This setup has been shown to cause unreadable intermediate steps in autoregressive models (e.g. DeepSeek-R1), and this is a reason why they have an additional SFT process. Despite this, attached examples in this paper show semantically meaningful and readable CoT like reasoning. I’m curious why the case is different with autoregressive models and outcome-based RL on diffusion language models that lead to interpretable intermediate reasoning.**
>
> Unlike auto-regressive models that are pretrained to predict next token from semantically meaningful prefix, DLMs are pretrained [a] to recover coherent text outputs from randomly corrupted inputs that are incomplete and unreadable with masked tokens. This training strategy endows DLMs with the unique robustness to generate semantically readable text outputs even from corrupted text sequences amid the reverse diffusion process. This trained ability of DLMs allows outcome-based RL to generate semantically meaningful reasoning texts without relying on additional SFT to recover the corrupted tokens during intermediate diffusion steps.
>
>
>
> [a] Nie S, et al. Large language diffusion models. 2025.

---

> > ### Author Response · Authors · 2025-08-06
> >
> > Dear Reviewer pRnQ，
> >
> > Thank you for your initial feedback on our work. We appreciate the time you’ve taken to evaluate our paper and have carefully addressed your questions in our rebuttal. To ensure all concerns are fully resolved before the discussion deadline (**Aug 8 AoE**), could you kindly let us know if you have any remaining concerns or need clarification on any points? We’re happy to provide additional details.

---

> > ### Comment · Reviewer_pRnQ · 2025-08-06
> >
> > Thank you for the detailed responses. I appreciate the additional analysis on generation length and the clarification regarding joint optimization across denoising steps, which helped address my initial concerns on those points.
> >
> > However, the explanation regarding interpretable intermediate reasoning under outcome-only RL remains largely qualitative. While the robustness of diffusion models is a plausible factor, this distinction from autoregressive models would benefit from deeper empirical or theoretical support to substantiate the claim. Given this, I am keeping my original score.

---

> ### Author Response · Authors · 2025-08-07
>
> Thanks for your feedback. Here we provide a quantitative evaluation on the robustness of the Diffusion Language Model (DLM) to substantiate this claim as requested by the reviewer.
>
> One of the key features of DLMs is the theoretical noise schedule applied for training - it trains the model to robustly recover coherent text outputs from randomly corrupted inputs at various noise levels, ensuring that the model can generate semantically readable text outputs at the end of the reverse diffusion process when the noise level reaches 0.
>
> In Tab. II, we use Qwen2.5-Math-PRM-7B [a] to evaluate the actual quality of the model outputs at each intermediate diffusion step, and compare it with the corresponding theoretical noise level. This aims to reveal the consistency of the actual quality with the theoretical noise level by the diffusion language model. For this, we report the average rewards at each step calculated on correctly answered GSM8K samples. During the reverse diffusion process, as the theoretical text cleanness (defined as 1 - noise level) increases from 0 to 1, the average reward steadily rises from 0.08 to 0.94 (very close to the maximum reward of 1.0). We also compute the correlation between the theoretical cleanness and the reward, reaching a high value of Pearson correlation 0.95.
>
> This verifies the noise levels structured in the DLM provides a consistent prediction on the quality of intermediate diffusion outputs, and the robustness of the DLM being able to recover the semantically meaningful reasonings comes from the scheduled various noise levels during the training and the resultant high consistency with the measured reward on the actual quality. This eventually allows the DLM to generate semantically meaningful text with increasing reward of quality as the theoretical noise level reaches 0 at the end of the reverse diffusion process.
>
> We hope this further analysis clears your concern and provides a quantitative analysis of DLM's robustness in generating semantically meaningful reasoning texts. Please let us know if you need more information.
>
> **Table II**: Relationship between theoretical cleanness (=1-noise level) and the average reward of actual quality on correctly answered GSM8K samples after our outcome-based RL training. The result shows a Pearson correlation as high as 0.95 between the noise levels and the measured rewards on the intermediate diffusion outputs.
>
>
> | Diffusion Step | Cleanness | Reward  |
> | ---- | -------- | ------------------- |
> | 0    | 0.00     | 0.08        |
> | 64   | 0.25     | 0.18        |
> | 128  | 0.50     | 0.35       |
> | 192  | 0.75     | 0.80        |
> | 256  | 1.00     | 0.94         |
>
>
> [a] Zhang, Z, et al, The Lessons of Developing Process Reward Models in Mathematical Reasoning

---

### Note · Authors · 2025-08-12

We thank the ACs and reviewers for valuable feedbacks. We have provided additional evidences reviewers requested, including:
- Comparing with d1 directly and applying DCoLT to Dream 7B. (Reviewer P9Mj)
- Quantitatively confirming that our model generates coherent text in the middle of the answer. (Reviewer pRnQ & P9Mj)
- Analyzing the impact of the generation length and the model performance across varying difficulties. (Reviewer pRnQ & h4x9)

Most of the reviewers responded that these rebuttals had clarified their concerns, except Reviewer pRnQ asked for additional support to further understand the **distinctive robustness of DLMs**. We provided **additional evidences in our further response**. While Reviewer pRnQ has yet to respond, we wish it will facilitate your discussions.

Moreover, we'd like to further clarify that the SFT mentioned by Reviewer pRnQ used in Deepseek-R1 to improve the readability of reasoning process actually contains two parts: 1) the **prior-RL SFT** based on the coldstart data before outcome-based RL, 2) the **additional post-RL SFT** based on both reasoning and non-reasoning data after outcome-based RL training [1]. However, unlike Deepseek-R1, most existing outcome-based RL works [2,3], including our own method, **only adopt** prior-RL SFT without additional post-RL SFT process. Even though, it has been demonstrated that these methods still generated **readable** reasoning steps without additional post-RL SFT [2,3].

Thus, for a fair comparison with them, we adopt the LLaDA as the base model that has also undergone prior-RL SFT with **no** post-RL one. As shown in Fig. 8&9 in the appendix, our model generated semantically meaningful text. In our response to Reviewer pRnQ, the measured reward of text quality in Tab. II is as high as 0.94 at the last diffusion step, which also confirms it from **a quantitative perspective**. Although post-RL SFT may further improve the readability of reasoning results, it was unnecessary and unfair to compare with other models [2,3].
Moreover it is straightforward to adopt post-RL SFT in DCoLT, but it is not our focus, and thus **unfair to ignore our main contributions** that Reviewer pRnQ also acknowledged to assess this paper.

[1] DeepSeek-R1: Incentivizing Reasoning Capability in LLMs via Reinforcement Learning

[2] DeepSeekMath: Pushing the Limits of Mathematical Reasoning in Open Language Models

[3] d1: Scaling Reasoning in Diffusion Large Language Models via Reinforcement Learning

---

### Decision · Program_Chairs · 2025-09-17

**Decision:**

Accept (poster)

**Comment:**

his paper presents diffusion chain of lateral thought, a method for enhancing reasoning ability of diffusion language models with reinforcement learning with outcome-reward. Two implementations for continuous and discrete time diffusion models are proposed respectively. Experiment results are strong on math, coding and sudoku benchmarks.

Strengths:
1. Novel ideas for bidirectional, non-linear reasoning.
2. Comprehensive experiments and strong results.
3. This paper is generally well-written.

Weaknesses:
1. some technical details are not fully clear. the authors should further revise their paper according to reviewers' comments.